# Enhanced flexoelectricity at reduced dimensions revealed by mechanically tunable quantum tunnelling

Saikat Das[1,2], Bo Wang [3], Tula R. Paudel [4], Sung Min Park[1,2], Evgeny Y. Tsymbal[4], Long-Qing Chen[3], Daesu Lee [5] & Tae Won Noh [1,2]

Flexoelectricity is a universal electromechanical coupling effect whereby all dielectric materials polarise in response to strain gradients. In particular, nanoscale flexoelectricity promises exotic phenomena and functions, but reliable characterisation methods are required to unlock its potential. Here, we report anomalous mechanical control of quantum tunnelling that allows for characterising nanoscale flexoelectricity. By applying strain gradients with an atomic force microscope tip, we systematically polarise an ultrathin film of otherwise non-polar $SrTiO_3$, and simultaneously measure tunnel current across it. The measured tunnel current exhibits critical behaviour as a function of strain gradients, which manifests large modification of tunnel barrier profiles via flexoelectricity. Further analysis of this critical behaviour reveals significantly enhanced flexocoupling strength in ultrathin $SrTiO_3$, compared to that in bulk, rendering flexoelectricity more potent at the nanoscale. Our study not only suggests possible applications exploiting dynamic mechanical control of quantum effect, but also paves the way to characterise nanoscale flexoelectricity.

---

[1] Center for Correlated Electron Systems, Institute for Basic Science (IBS), Seoul 08826, Korea. [2] Department of Physics and Astronomy, Seoul National University, Seoul 08826, Korea. [3] Department of Materials Science and Engineering, Pennsylvania State University, University Park, Pennsylvania 16802, USA. [4] Department of Physics and Astronomy & Nebraska Center for Materials and Nanoscience, University of Nebraska, Lincoln, Nebraska 68588, USA. [5] Department of Physics, Pohang University of Science and Technology (POSTECH), Pohang 37673, Korea. Correspondence and requests for materials should be addressed to D.L. (email: dlee1@postech.ac.kr) or to T.W.N. (email: twnoh@snu.ac.kr)

Polar materials form the basis of electromechanics, optoelectronics and studies on emerging quantum states[1]. Such materials belong to only 10 of the 32 possible crystal point groups, and sometimes exhibit problematic size effects[2]. Under such circumstances, flexoelectricity[3–5] offers unique advantages[6–15]. Strain gradients can intrinsically polarise all materials with arbitrary crystal symmetries[3–5], ranging from dielectrics[16] to semiconductors[11] and from bio-materials[17] to two-dimensional materials. Importantly, such ubiquitous flexoelectric effects potentially become even larger at the nanoscale, as strain gradients scale inversely with material size. Nanoscale strain-graded dielectrics (e.g. a strain variation $\Delta u = 1\%$ within 1 nm) encompass enormous strain gradients (to $\partial u/\partial x = 10^7\,\mathrm{m^{-1}}$) and may exhibit remarkable phenomena and flexoelectric functionality[6–15]. Furthermore, nanoscale flexoelectricity can fundamentally differ from the conventional bulk flexoelectricity, e.g. due to a nonlinear polarisation response under large strain gradients[9]. Thus, characterising nanoscale flexoelectricity is of great importance from both a fundamental and technological viewpoint.

For characterising nanoscale flexoelectricity, it is necessary to identify a nanoscale phenomenon that can be actively controlled by the flexoelectric effect. It is well established that the quantum tunnelling probability through a nanometre-thick ferroelectric barrier layer sandwiched between two metallic electrodes sensitively depends on the polarisation direction and its magnitude[18–20]. In this so-called ferroelectric tunnel device, the depolarisation field, originating from the imperfect screening of ferroelectric polarisation by the metallic electrodes, alters the intrinsic barrier height. For asymmetric electrodes, changing the polarisation direction yields two different effective barrier heights, and subsequently leads to two discrete electroresistance states. Meanwhile, due to the converse piezoelectric effect, the barrier width can also modulate in response to the electric-field applied during the tunnelling transport measurement[21–25]. This also leads to dissimilar electroresistance states. All these considerations suggest a possibility of controlling quantum tunnelling via flexoelectric effect, thereby allowing for characterisation of nanoscale flexoelectricity.

Here, we demonstrate that a systematic control of quantum tunnelling through a paraelectric ultrathin SrTiO₃ (STO) film by flexoelectric polarisation allows characterising nanoscale flexoelectricity. By applying the strain gradients from a conductive scanning probe tip, we simultaneously polarise and measure the tunnelling current across the film. With increasing strain gradients, the tunnelling current exhibits an asymmetric–symmetric crossover, which we attribute, based on the Wentzel–Kramers–Brillouin (WKB) modelling, to flexoelectric polarisation-driven modification of the tunnelling barrier profile. Furthermore, analysing the modification of the barrier profile as a function of strain gradients enables quantifying the flexocoupling coefficient, which we find becomes much enhanced compared to the bulk value. We discuss possible origins of this enhanced flexocoupling coefficient.

## Results

**Concept of flexoelectric control of quantum tunnelling**. Figure 1a shows a schematic of our experimental setup. We use a conductive atomic force microscope (AFM) tip (PtIr-coated) to apply strain gradients[8] and simultaneously measure the tunnel current. We systematically generate giant strain gradients (up to $>10^7\,\mathrm{m^{-1}}$), as estimated by contact mechanics analysis (Fig. 1b and Methods). These strain gradients are much larger than those achievable using a conventional beam-bending approach, which generates gradients in the range of $10^{-1}\,\mathrm{m^{-1}}$ (using micrometre-thick beams)[16] to $10^2\,\mathrm{m^{-1}}$ (employing nanometre-thick beams)[10]. When an ultrathin dielectric layer becomes flexoelectrically polarised by a giant strain gradient, the resulting depolarisation field and electrostatic contribution[18–20] significantly modify the tunnel barrier profile (Fig. 2a–c). Therefore, we can utilize pure mechanical force by an AFM tip as a dynamic tool not only for systematically controlling quantum tunnelling, but also for characterising nanoscale flexoelectricity.

As a model system, we choose the archetypal dielectric material SrTiO₃ (STO), which remains paraelectric down to a temperature of 0 K in bulk. We prepare homoepitaxial ultrathin STO films on (001)-oriented STO substrates with a conductive SrRuO₃ (SRO) buffer layer. To avoid the off-stoichiometry-driven ferroelectric phase of STO (ref. [26]), we use an ultra-slow growth scheme[27], combined with in situ post-annealing in oxygen to minimize oxygen vacancies. Piezoresponse force microscopy confirms that the STO films are indeed paraelectric (Supplementary Fig. 1). Notably, our geometry induces compressive strains in both the

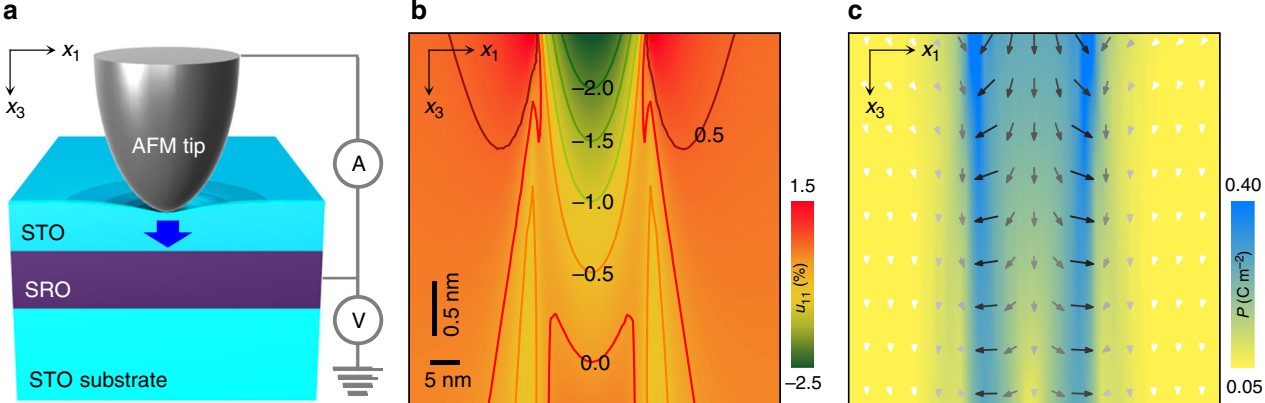

**Fig. 1** Electron tunnelling through a flexoelectrically polarised ultrathin barrier. **a** Schematic of the experimental setup, illustrating flexoelectric polarisation (blue arrow) generated by the atomic force microscope (AFM) tip pressing the surface of ultrathin SrTiO₃ (STO). **b** Simulated transverse strain $u_{11}$ in a nine unit cell-thick (i.e. 3.5 nm-thick) STO under a representative tip loading force of 5 μN. Along the central line, $u_{11}$ varies by ~0.5% within $\Delta x_3 = 0.5$ nm, yielding $\partial u_{11}/\partial x_3 \sim 10^7\,\mathrm{m^{-1}}$. **c** Polarisation profile, obtained by phase-field simulation, for the strain profile in **b**. Arrows denote the polarisation direction. In the tip-contact region, the polarisation along the $x_3$ direction was around 0.17 C m⁻² on average. Note that when neglecting flexoelectricity (i.e. $f = 0$), our simulation does not produce any polarisation in STO, which again confirms the flexoelectricity-based origin of our observation. Source data are provided as a Source Data file

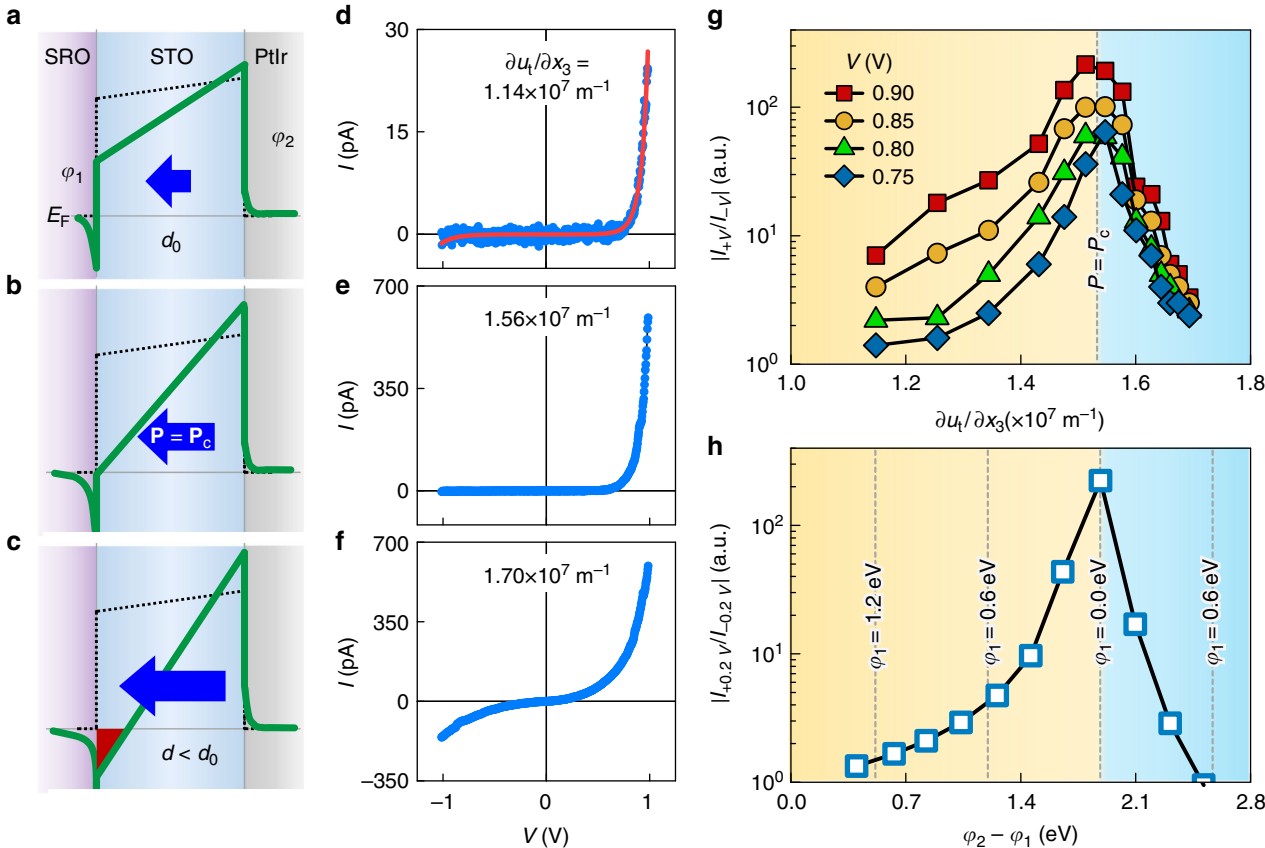

**Fig. 2** Flexoelectric control of electron tunnelling. **a–c** Schematics of the potential energy profiles across SrTiO₃ (STO) with increasing flexoelectric polarisation (**P**; blue arrows). The additional electrostatic potential induced by $P$ modifies the original barrier potential energy (black dotted line) to yield the total potential energy (green solid line). At the critical polarisation $P_c$, the tunnel barrier becomes triangular with $\varphi_1 = 0$ and $\varphi_2 = \varphi_{0,2} + \varphi_{0,1} \cdot (\delta_{PtIr}/\delta_{SRO})$. **d–f** Measured tunnel current–voltage ($I$–$V$) curves across the nine unit cell-thick STO film for three representative $\partial u_t/\partial x_3$ values. The red solid line in **d** indicates the fit to Equation (5). **g** The rectification ratios (RRs) $|I_{+V}/I_{-V}|$ of the measured tunnel current as a function of $\partial u_t/\partial x_3$. With increasing $\partial u_t/\partial x_3$, the tunnelling $I$–$V$ curves become more asymmetric in regime (A) (yellow) below $\partial u_t/\partial x_3 = 1.56 \times 10^7$ m⁻¹, but more symmetric in regime (B) (blue). **h** The simulated $|I_{+V}/I_{-V}|$ at $V = 0.2$ V as a function of barrier-asymmetry, defined as $\varphi_2 - \varphi_1$. We set the initial barrier heights as $\varphi_1 = 1.3$ eV and $\varphi_2 = 1.7$ eV, and systematically decrease $\varphi_1$ (or increased $\varphi_2$) while fulfilling the condition $(1.3 - \varphi_1)/(\varphi_2-1.7) = 8$. Source data are provided as a Source Data file

transverse $x_1$ and longitudinal $x_3$ directions (Fig. 1b and Supplementary Fig. 4), attributable to AFM tip-induced downward bending and pressing. Such three-dimensional compression of STO does not favour either ferroelectricity or piezoelectricity. This makes it possible to explore pure flexoelectric polarisation, the electrostatic effect of which modifies the STO tunnel barrier[18,19].

**Strain-gradient-dependent tunnelling transport.** We measure the tunnel current across a nine unit cell-thick (i.e. ~3.5 nm-thick) STO as a function of the applied strain gradients. Our theoretical analysis reveals that the transverse strain gradients defined as

$$\partial u_t/\partial x_3 = \partial u_{11}/\partial x_3 + \partial u_{22}/\partial x_3 \qquad (1)$$

are an order of magnitude larger than the longitudinal strain gradients $\partial u_{33}/\partial x_3$ (Supplementary Note 2 and Supplementary Fig. 4); we thus consider only the $\partial u_t/\partial x_3$ values. Figure 2d–f show the measured current–voltage ($I$–$V$) curves for three representative $\partial u_t/\partial x_3$ values (see Supplementary Fig. 2 for the entire set). For $\partial u_t/\partial x_3 < 1.56 \times 10^7$ m⁻¹ (Fig. 2d), the $I$–$V$ curves exhibit typical tunnelling characteristics (red solid line) and are highly asymmetric, manifesting rectifying behaviour. The forward current systematically increases with increasing $\partial u_t/\partial x_3$, whereas

the reverse current remains comparable to the noise level (~1 pA). When $\partial u_t/\partial x_3$ attains a critical value, $1.56 \times 10^7$ m⁻¹, the $I$–$V$ curve became maximally asymmetric (Fig. 2e). Beyond a $\partial u_t/\partial x_3$ of $1.56 \times 10^7$ m⁻¹, however, the reverse current begins to increase, whereas the forward current increases only marginally, rendering the $I$–$V$ curve more symmetric (Fig. 2f). Figure 2g emphasizes this critical behaviour by plotting rectification ratios (RR $\equiv |I_{+V}/I_{-V}|$) as a function of $\partial u_t/\partial x_3$. We also observe a similar critical behaviour in an eleven unit cell-thick STO film (Supplementary Fig. 3).

Before addressing how flexoelectricity could explain these results, we rule out other possible origins of the phenomena. First, the AFM tip-induced pressure does not cause any permanent surface damage to the STO film (Supplementary Fig. 5). Additionally, the mechanical control of electron tunnelling is reversible (Supplementary Fig. 6), excluding any involvement of an electrochemical process. We also consider the effect of strain on the STO tunnel barrier profiles. AFM tip-induced compressive strain per se would not only decrease the barrier width ($\Delta d \leq 0.2$ nm) but also slightly increase the STO band gap[28] and hence the barrier height. However, our detailed analysis show that the strain effect is too small to explain our observations (Supplementary Note 3). Furthermore, we confirm that the strain-induced changes in electronic properties of SRO are too small to be responsible for the anomalous behaviour of tunnelling transport

(Supplementary Note 6). Thus, the asymmetric–symmetric crossover is an intrinsic effect possibly attributable to flexoelectric polarisation-induced modification of the tunnel barrier.

**Understanding and modelling the tunnelling transport.** To understand how the barrier profile affects tunnel current, we perform a one-dimensional WKB simulation of a metal-insulator-metal (M1-I-M2) heterostructure (Supplementary Note 4). In the experiment, 'M1', 'M2', and 'I' correspond to SRO, PtIr, and STO, respectively. Our calculations suggest that the observed rectifying tunnelling behaviour should originate from an asymmetric, trapezoidal barrier profile, with the barrier height $\varphi_1$ at the M1-I interface being smaller than the barrier height $\varphi_2$ at the I-M2 interface (as in Fig. 2a). Such an asymmetric tunnel barrier implies downward flexoelectric polarisation (pointing towards the M1/I interface), and a higher probability of transmission to the M1 electrode than in the reverse direction (to the M2 electrode). When flexoelectric polarisation attains a critical value, the tunnel barrier becomes triangular, such that $\varphi_1 = 0$ (as in Fig. 2b), yielding the maximum rectifying behaviour. This explains the anomalous increase in RR in regime (A) (yellow) of Fig. 2h.

When the flexoelectric polarisation increases further, the conduction band minimum of STO could cross the Fermi level (as in Fig. 2c). This crossing metallizes the interfacial barrier layer and concomitantly decreases the effective barrier width $d$, as supported by first-principles calculations (Fig. 3, Supplementary Note 5 and Methods). For convenience, we describe this case using a negative $\varphi_1$ (Supplementary Fig. 10). As the RR of a triangular tunnel barrier is exponentially proportional to the barrier width $d$, the decrease in $d$ would lower the RR, as shown in regime (B) (blue) of Fig. 2h. Notably, the barrier-asymmetry dependence of RR (Fig. 2h) strikingly resembles the experimentally observed strain-gradient dependence of $RR$ (Fig. 2g); both exhibit the asymmetric–symmetric crossover. Therefore, we conclude that flexoelectric polarisation-induced metallization near the SRO/STO interface manifests itself as an asymmetric–symmetric crossover in tunnelling transport.

Next, to understand how the barrier profile varies with increasing $\partial u_t/\partial x_3$, we fit the tunnel spectra of regime (A) to an analytical equation[20] that describes tunnelling through a trapezoidal barrier (red solid line in Fig. 2d; see Methods and Supplementary Note 1). Taking into account the work functions of SRO (5.2 eV)[29] and PtIr (5.6 eV)[30], and the electron affinity of STO (3.9 eV)[29], we set the intrinsic barrier heights $\varphi_{0,1}$ and $\varphi_{0,2}$ to 1.3 and 1.7 eV, respectively (black dotted line in Fig. 2a). Furthermore, following simple electrostatics argument[18,19], we constrain the $\varphi_1$ and $\varphi_2$ to vary obeying the relation:

$$\Delta\varphi_1/\Delta\varphi_2 = \left(\varphi_{0,1} - \varphi_1\right)/\left(\varphi_2 - \varphi_{0,2}\right) = \delta_{\text{SRO}}/\delta_{\text{PtIr}}, \quad (2)$$

where $\delta_{\text{SRO}}$ and $\delta_{\text{PtIr}}$ are the effective screening lengths of SRO and PtIr, respectively. Given that $\delta_{\text{SRO}} \approx 0.5$–0.6 nm (ref. [2]) and $\delta_{\text{PtIr}} < 0.1$ nm, we set $\Delta\varphi_1/\Delta\varphi_2$ ($= \delta_{\text{SRO}}/\delta_{\text{PtIr}}$) to be 8. Figure 4a plots the fitted $\varphi_1$ and $\varphi_2$ values as a function of $\partial u_t/\partial x_3$. Consistent with the WKB simulations, our fitting yields highly asymmetric trapezoidal barrier profiles, where with increasing $\partial u_t/\partial x_3$, $\varphi_1$ decreases from 0.57 to 0.34 eV, and $\varphi_2$ increases from 1.79 to 1.82 eV.

**Quantifying flexocoupling coefficient at the nanoscale.** Based on this $\partial u_t/\partial x_3$ dependence of $\varphi_1$ and $\varphi_2$, we estimate the strength of effective flexoelectric coupling. The transverse strain gradient polarises the STO layer through the flexoelectric effect, and the

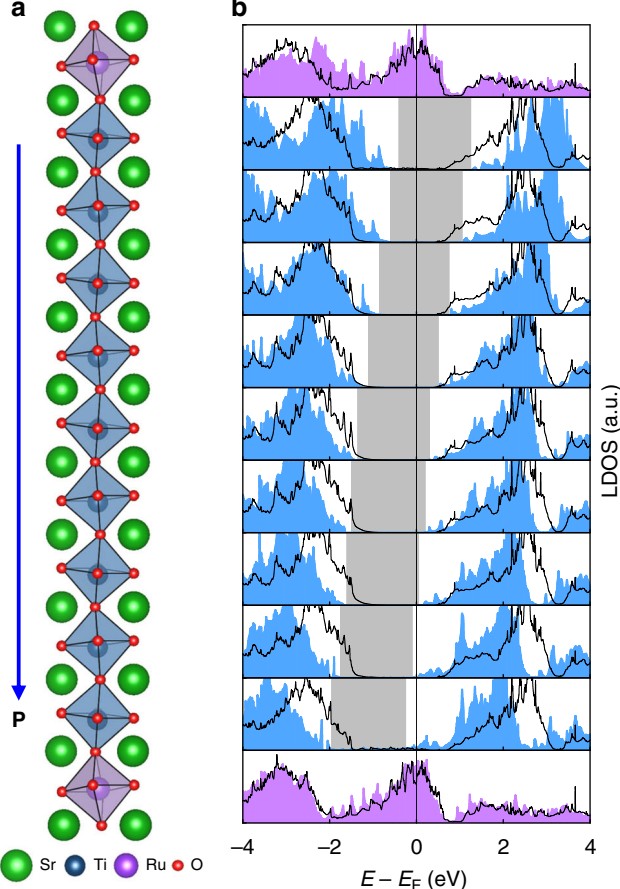

**Fig. 3** Polarisation-induced local metallization in SrTiO₃. **a** The simulation cell. We artificially polarise SrTiO₃ (STO) layers with uniform displacement of Ti atom by 0.2 Å. **b** Calculated layer-resolved density of states (LDOS) of polarised STO layers (filled blue) compared to that of nonpolar STO (black solid line). Grey regions represent a gap between conduction band minimum and valence band maximum of polarised STO layers, clearly showing a shift of energy bands due to polarisation-induced electric field. Source data are provided as a Source Data file

induced polarisation can be expressed as

$$P = \mu_{\text{eff}} \cdot (\partial u_t/\partial x_3) = \varepsilon \cdot f_{\text{eff}} \cdot (\partial u_t/\partial x_3) \quad (3)$$

where $\mu_{\text{eff}}$, $\varepsilon$ and $f_{\text{eff}}$ are the effective flexoelectric coefficient, the dielectric permittivity and the effective flexocoupling coefficient of STO, respectively. In ultrathin STO, this flexoelectric polarisation results in depolarisation field ($\propto -P/\varepsilon$) and modifies the tunnel barrier profile according to the following electrostatic equation (Supplementary Note 7):[18,19]

$$\left(\varphi_2 - \varphi_1\right)/ed = P/\varepsilon + E_{\text{bi}} = f_{\text{eff}} \cdot (\partial u_t/\partial x_3) + E_{\text{bi}}, \quad (4)$$

where $e$ is the electronic charge and $E_{\text{bi}}$ is the additional built-in field contribution that could arise from the work function difference between SRO and PtIr, surface dipoles[31], and/or an offset between the calculated and actual strain gradients. As shown in Fig. 4b, the calculated $(\varphi_2 - \varphi_1)/ed$ varies almost linearly with $\partial u_t/\partial x_3$ (grey solid line), giving a slope $f_{\text{eff}}$ of $23 \pm 1$ V. In addition, fitting also yields a nonzero contribution at $\partial u_t/\partial x_3 = 0$ (i.e. $8$–$10 \times 10^7$ V m⁻¹), corresponding to the built-in field $E_{\text{bi}}$.

We now focus on the onset of asymmetric–symmetric crossover of tunnel current at $(\partial u_t/\partial x_3)_c = 1.56 \times 10^7$ m⁻¹, which also allows us to estimate $f_{\text{eff}}$. According to our simulation, this crossover is attributable to the polarisation-induced trapezoidal-

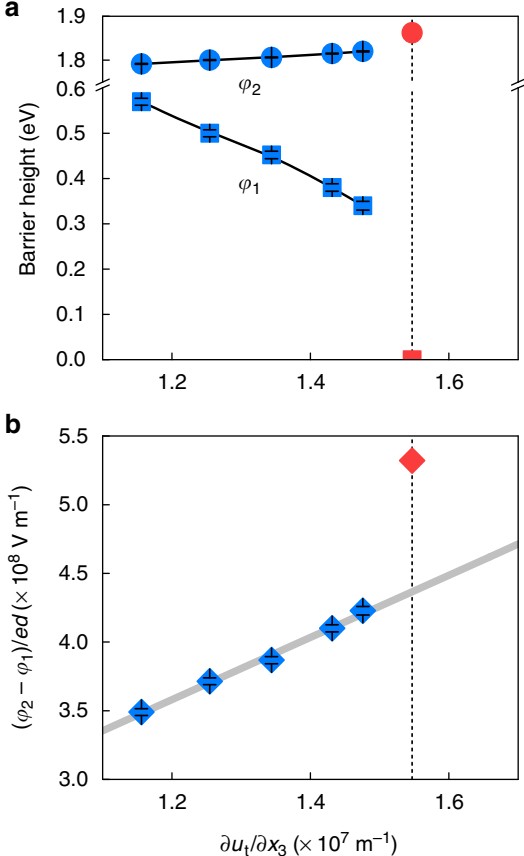

**Fig. 4** Characterising flexoelectricity in ultrathin SrTiO₃. **a** The blue squares and circles indicate $\varphi_1$ and $\varphi_2$, respectively, extracted by fitting the tunnelling spectra to Equation (5). The error bars represent the standard deviations of the extracted barrier heights. The red square and circle represent $\varphi_1$ and $\varphi_2$, respectively, for the triangular barrier at the critical $\partial u_t / \partial x_3$. **b** $(\varphi_2 - \varphi_1)/ed$, calculated from **a**. The grey line shows a linear fit to the data. The error bars represent the standard deviations. Source data are provided as a Source Data file

to-triangular transition of the tunnel barrier. At the critical $\partial u_t / \partial x_3$ (or equivalently, at the critical $P$), we therefore expect that $\varphi_1 = 0$ and $\varphi_2 = \varphi_{0,2} + \varphi_{0,1} \cdot (\delta_{PtIr}/\delta_{SRO}) = 1.7 + 1.3/8$ eV, giving $(\varphi_2 - \varphi_1)/ed = 5.32 \times 10^8$ V m⁻¹ (Fig. 4). With $E_{bi} = 9 \times 10^7$ V m⁻¹ and $(\partial u_t / \partial x_3)_c = 1.56 \times 10^7$ m⁻¹, Equation (4) yields $f_{eff} = 28$ V. This value compares reasonably well to that ($23 \pm 1$ V) obtained from fitting, demonstrating that our approach is innately consistent. Furthermore, using the obtained $f_{eff}$, we simulate a three-dimensional profile of local $P$ (Fig. 1c and Methods), and find that the average out-of-plane $P$ is around 0.17 C m⁻² for $\partial u_t / \partial x_3 = 1.6 \times 10^7$ m⁻¹ (i.e. just above the critical $\partial u_t / \partial x_3$). This value compares well to the predicted critical polarisation (i.e. $P_c = 0.16$ C m⁻²; Supplementary Note 8), which again emphasizes the self-consistency of our approach.

## Discussion

Interestingly, the estimated flexocoupling coefficient (23–28 V) is larger than Kogan's phenomenological estimate (1–10 V)[3,5], and indeed an order of magnitude greater than the experimental value (~2.6 V) for bulk STO (ref.[16]). To understand this enhancement, we first note that a nonlinear flexoelectric response could arise under large strain gradients, as demonstrated in several material systems[9,32]. By considering the nonlinear flexoelectricity, e.g. third-order response (Supplementary Note 9), we might explain

the enhancement of $f_{eff}$ under a huge $\partial u_t / \partial x_3$. Additionally, a surface contribution $f_{surf}$ can be involved[33,34], which, combined with the bulk contribution $f_{bulk}$, determines the overall coupling coefficient $f_{eff}$ ($= f_{surf} + f_{bulk}$) of a material. When considered separately, both $f_{surf}$ and $f_{bulk}$ could be $> 10$ V in magnitude but opposite in sign[35]. Our results may thus imply that only either the surface or bulk contribution becomes dominant in the ultrathin limit. To obtain a complete understanding of enhanced flexoelectricity in ultrathin STO, further systematic experimental and theoretical investigations will be required.

In summary, we show that quantum tunnelling is mechanically tunable. Such mechanical tunability allows experimentally determining the flexocoupling strength at the nanoscale, which we find to be much enhanced compared to that in bulk. This finding emphasizes that flexoelectricity could become much more powerful at reduced dimensions due to not only a large strain gradient but also an enhanced coupling strength. We hope that this study would encourage the construction of flexocoupling coefficient databases at the nanoscale, and the design of high-performance flexoelectric devices. From a broader perspective, this study highlights several favourable aspects of nanoscale flexoelectricity. First, nanoscale flexoelectricity allows for the generation of large polarisation in a continuous manner. We start from a nonpolar STO and continuously polarise it up to a polarisation value of 0.4 C m⁻². Second, such a continuously tunable, large polarisation can also generate a large electrostatic potential, which corresponds to a stationary effective electric field, as high as $10^9$ V m⁻¹. This can be useful for a large electric-field control of dielectrics, which has been challenging due to dielectric breakdown.

## Methods

**Sample fabrication.** SRO and STO thin films were sequentially grown on TiO₂-terminated and (100)-oriented STO substrates. The growth dynamics and thicknesses were monitored by in situ reflection high-energy electron diffraction (RHEED). Film deposition was performed at 700 °C under oxygen partial pressures of 100 and 7 mTorr for SRO and STO, respectively. After deposition, films were annealed at 475 °C for 1 h in oxygen at ambient pressure and subsequently cooled to room temperature at 50 °C min⁻¹. Structural characterisation, namely, the reciprocal space mapping was performed to ensure that the STO film is strain-free (Supplementary Fig. 14).

**Tunnelling measurements.** Current–voltage curves were obtained using an Asylum Research Cypher AFM at room temperature under ambient conditions. Conducting PtIr-coated metallic tips (NANOSENSORS™ PPP-EFM) with nominal spring constants 50–60 N m⁻¹, and a dual-gain ORCA module, were used to measure currents. An electrical bias was applied through the conducting SRO electrode; this was swiped from -1 V to +1 V at a ramping rate of about 4 V s⁻¹. The noise floor of the AFM system was about ~1 pA.

To extract barrier heights from the tunnelling I–V curves, we used an analytical equation describing direct tunnelling through trapezoidal tunnel barriers:[20,36]

$$I(V) \cong b + c \frac{\exp\left\{\alpha(V)\left[(\varphi_2 - \frac{eV}{2})^{\frac{3}{2}} - (\varphi_1 + \frac{eV}{2})^{\frac{3}{2}}\right]\right\}}{\alpha^2(V)\left[(\varphi_2 - \frac{eV}{2})^{\frac{1}{2}} - (\varphi_1 + \frac{eV}{2})^{\frac{1}{2}}\right]^2} \tag{5}$$

$$\sinh\left\{\frac{3}{2}\alpha(V)\left[(\varphi_2 - \frac{eV}{2})^{\frac{1}{2}} - (\varphi_1 + \frac{eV}{2})^{\frac{1}{2}}\right]\frac{eV}{2}\right\}$$

where $c$ is a constant and $\alpha(V) \equiv [4d(2m_e)^{1/2}]/[3\hbar(\varphi_1 + eV - \varphi_2)]$. Also, $b$, $m_e$, $d$, and $\varphi_{1,2}$ are the baseline, free electron mass, barrier width, and barrier height, respectively. As explained in the main text, our fittings imposed the constraints $\varphi_2 = 1.7 + \Delta\varphi$ and $\varphi_1 = 1.3 - 8\Delta\varphi$. In addition, we used a scaling factor to account for the increase in contact area with increasing contact force, but this did not affect our principal results (i.e. the RRs, $|I_{+V}/I_{-V}|$). For smaller $\Delta\varphi$ values, we used the entire tunnelling spectra for fitting (Supplementary Fig. 2a–c). However, when larger distortions of the barrier profiles were apparent (i.e. at larger $\Delta\varphi$ values), we fitted the tunnelling spectra using smaller bias windows.

**Simulation of strain profile.** The strain distribution in a 3.5 nm-thick STO thin film pressed with an AFM tip is obtained by solving the elastic equilibrium equation by using Khachaturyan microelasticity theory[37] and the Stroh formalism of anisotropic elasticity[38]. The detailed procedure has been elaborated in previous

works[39]. Here, we discretized three-dimensional space into $64 \times 64 \times 700$ grid points and applied periodic boundary conditions along the $x_1$ and $x_2$ axes. The grid spacing was $\Delta x_1 = \Delta x_2 = 1$ nm and $\Delta x_3 = 0.1$ nm. Along the $x_3$ direction, 35 layers were used to mimic the film; the relaxation depth of the substrate featured 640 layers to ensure that the displacement at the bottom of substrate was negligibly small. To estimate surface stress distribution that developed on AFM-tip pressing, we adopted the Hertz contact mechanics of the spherical indenter[40] with a tip radius of 30 nm and a mechanical force of 1–7 μN. The Young's moduli and Poisson ratios of the Pt tip and the STO film were $E^{\text{tip}} = 168$ GPa and $v^{\text{tip}} = 0.38$, and $E^{\text{film}} = 264$ GPa and $v^{\text{film}} = 0.24$, adapted from ref. [41]. The electrostrictive and rotostrictive coupling coefficients of STO were adapted from ref. [42]. See Supplementary Note 2 for more details.

**Simulation of polarisation profile.** The polarisation distribution under the mechanical load by an AFM tip can be calculated by self-consistent phase-field modelling[43]. The temporal evolution of polarisation field $\mathbf{P}(\mathbf{x},t)$ is governed by the time-dependent Ginzburg–Landau equation, i.e. $\partial \mathbf{P}/\partial t = -L(\delta F(\mathbf{P})/\delta \mathbf{P})$, where $L$ is the kinetic coefficient. The total free energy $F$ can be expressed as[43]

$$F = \int (f_{\text{bulk}} + f_{\text{elastic}} + f_{\text{electric}} + f_{\text{gradient}} + f_{\text{flexo}}) dV$$

$$= \int \left[ \begin{array}{l} \alpha_{ij}P_iP_j + \alpha_{ijkl}P_iP_jP_kP_l + \beta_{ij}\theta_i\theta_j + \beta_{ijkl}\theta_i\theta_j\theta_k\theta_l + t_{ijkl}P_iP_j\theta_k\theta_l + \frac{1}{2}g_{ijkl}\frac{\partial P_i}{\partial x_j}\frac{\partial P_k}{\partial x_l} \\ + \frac{1}{2}k_{ijkl}\frac{\partial \theta_i}{\partial x_j}\frac{\partial \theta_k}{\partial x_l} + \frac{1}{2}c_{ijkl}\left(\varepsilon_{ij} - \varepsilon_{ij}^0\right)\left(\varepsilon_{kl} - \varepsilon_{kl}^0\right) - \frac{1}{2}E_iP_i + \frac{1}{2}f_{ijkl}\left(\frac{\partial P_k}{\partial x_l}\varepsilon_{ij} - \frac{\partial \varepsilon_{ij}}{\partial x_l}P_k\right) \end{array} \right] dV$$

$$(6)$$

The bulk Landau free energy $f_{\text{bulk}}$ consists of two sets of order parameters, i.e. the spontaneous polarisation $\mathbf{P}$ and the antiferrodistortive order parameter $\mathbf{\theta}$, which represents the oxygen octahedral rotation angle of STO (ref. [42]). The flexoelectric contribution is considered as a Liftshitz invariant term as

$$f_{\text{flexo}} = \frac{1}{2}f_{ijkl}\left(\frac{\partial P_k}{\partial x_l}\varepsilon_{ij} - \frac{\partial \varepsilon_{ij}}{\partial x_l}P_k\right) \quad (7)$$

The eigenstrain tensor $\boldsymbol{\varepsilon}^0$ in the elastic energy density is given by

$$\varepsilon_{ij}^0 = Q_{ijkl}P_kP_l + \Lambda_{ijkl}\theta_k\theta_l - F_{ijkl}P_{k,l} \quad (8)$$

where the electrostrictive, rotostrictive and converse flexoelectric couplings are considered via tensor $\mathbf{Q}$, $\mathbf{\Lambda}$ and $\mathbf{F}$. The coefficients used in constructing the total free energy $F$ of STO single crystal were given in our previous works[42,44]. The transverse flexoelectric constant of STO estimated from experiments in present work were used ($f_{12} = 25$ V), while the other two flexoelectric component are assumed to be zero (i.e. $f_{11} = f_{44} = 0$) for simplicity.

**First-principles calculations.** The atomic and electronic structure of the system was obtained using the density functional theory (DFT) implemented in the Vienna ab initio simulation package (VASP)[45,46]. The projected augmented plane wave (PAW) method was used to approximate the electron-ion potential[47]. The exchange and correlation potentials were calculated using the local spin density approximation (LSDA). In calculation, we employed a kinetic energy cutoff of 340 eV for PAW expansion, and a $6 \times 6 \times 1$ grid of $\mathbf{k}$ points[48] for Brillouin zone integration. The in-plane lattice constant was that of relaxed bulk STO ($a = 3.86$Å); the $c/a$ ratio and the internal atomic coordinates were relaxed until the Hellman–Feynman force on each atom fell below $|0.01|$ eV Å$^{-1}$. The dielectric constant were calculated using density functional perturbation theory[49–51]. See Supplementary Note 5 for more details.

## Data availability
All relevant data presented in this manuscript are available from the authors upon reasonable request. The source data underlying Fig. 1–4 and Supplementary Fig. 1–4, 6 and 14 are provided as a Source Data file.

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

## Acknowledgements

This work was supported by the Research Center Program of the IBS (Institute for Basic Science) in Korea (grant no. IBS-R009-D1). D.L. acknowledges the support by the National Research Foundation of Korea (NRF) grant funded by the Korea government (MSIT) (No. 2018R1A5A6075964). B.W. acknowledges the support by the NSF-MRSEC grant number DMR-1420620. The effort of L.-Q.C. is supported by National Science Foundation (NSF) through Grant No. DMR-1744213. The work at the Pennsylvania State University used the Extreme Science and Engineering Discovery Environment (XSEDE) Bridges at the Pittsburgh Supercomputing Center through allocation TG-DMR170006, which is supported by National Science Foundation grant number ACI-1548562[52]. The research at the University of Nebraska−Lincoln is supported by the National Science Foundation through the Nebraska Materials Research Science and Engineering Center (MRSEC), Grant No. DMR-1420645.

## Author Contributions

D.L. conceived and designed the research. D.L. and T.W.N. directed the project. S.D. fabricated thin films and measured tunnelling transport assisted by S.M.P. and B.W. carried out simulations of strain gradient and polarisation under the supervision of L.-Q.C., T.R.P. and E.Y.T. carried out first-principles calculations. S.D. and D.L. prepared the manuscript. All authors discussed the results and implications and commented on the manuscript at all stages.

## Additional information

**Competing interests:** The authors declare no competing interests.

**Journal Peer Review Information**: *Nature Communications* thanks the anonymous reviewers for their contribution to the peer review of this work. Peer reviewer reports are available.

