## [Peer Review File · Nature Communications]

Reviewers' comments:

Reviewer #1 (Remarks to the Author):

Review report on: s41598-018-27616-6 (1)

Enhanced flexoelectricity at reduced dimensions revealed by mechanically tunable quantum tunnelling by Saikat Das et al.

The authors present an interesting study by combining flexoelectricity and quantum mechanical tunneling. Using a conductive AFM (atomic force microscope) the flexoelectric effect is induced and simultaneously the implication detected by the measured tunneling current.

The experimental and theoretical issues have been carefully and thoughtfully done. Concerning mechanical related effects and quantum mechanical tunneling the authors overlooked the literature on the piezoelectric effect and electron transport in FTJs. Because the work the author present its novel, its necessary to include in the introduction also a paragraph and the piezo-effect on electron tunneling to complete the introduction. Here a number of references which should be cited?

Converse-piezoelectric effect on current-voltage characteristics of symmetric ferroelectric tunnel junctions. Xiaoyan Lu, Wenwu Cao, Wenhua Jiang, and Hui Li J. Appl. Phys. 111, 014103 (2012); doi: 10.1063/1.3673600

Electroresistance Effect in Ferroelectric Tunnel Junctions with Symmetric Electrodes, Daniel I. Bilc et al., ASC Nano 2012

A ferroelectric tunnel junction based on the piezoelectric effect for non-volatile nanoferroelectric devices, Shuoguo Yuan et al., J. Mater. Chem. C, 2013, 1, 418

Theoretical current-voltage characteristics of ferroelectric tunnel junctions . H. Kohlstedt, N. A. Pertsev, J. Rodríguez Contreras,¹ and R. Waser, Phys. Rev. B 72, 125341 (2005)

Giant piezoelectric resistance in ferroelectric tunnel junctions . Yue Zheng¹ and C H Woo . Nanotechnology 20 (2009) 075401

In the supplements the authors confirm a stress free STO film by PFM. It would be more convincing to include a XRD (reciprocal space mapping) as well.

The effect on electron tunneling is attributed to the flexoelectric effect. The effect on strain on the SrRuO₃ was not taken into account. Can the authors indeed exclude any effect on the band structure of the SRO electrode due to the induced strain gradient? This might lead to a modified tunneling current.

By calculating the I(V) characteristics the authors the electron mass m_e (see equation S1/suppl. material). It is unclear whether the authors use the free electron mass or an effective mass. This issue should be clarified.

Reviewer #2 (Remarks to the Author):

This manuscript by Saikat Das et al provides an really important method to characterize nanoscale flexoelectricity, and it is extremely important since strain gradient will much stronger than that in macroscopic materials.

Minor points:

The strain gradient is theoretical analyzed by the Hertz contact mechanics of the spherical indenter for elasticity or anisotropic materials, whether it is valid for flexoelectric materials need to be discussed.

The transverse strain gradient $\partial u_t / \partial x_3 = \partial u_{11} / \partial x_3 + \partial u_{33} / \partial x_3$ is considered to represent the values of strain gradient, this may not correct since when the P_3 component is measured in the experiments. There are only three independent flexoelectric coefficient for cubic SrTiO3 materials, and $P_3 = \mu_{1133} \frac{\partial u_{11}}{\partial x_3} + \mu_{3333} \frac{\partial u_{33}}{\partial x_3}$, the transverse strain gradient $\partial u_t / \partial x_3$ cannot represent the combination contribution of the flexoelectric effect.

Flexoelectric effect will generate a built-in electric field and it can be represented as $E_{flexo} = f_{eff} \partial u_t / \partial x_3$, therefore, Eq. (1) may be rewritten as

$$(\varphi_2 - \varphi_1) / ed = E_{flexo} + E_{bi} = f_{eff} (\partial u_t / \partial x_3) + E_{bi}$$

The electric polarization induced by the strain gradient is $P_{flexo} = \mu_{eff} (\partial u_t / \partial x_3)$, and when only the flexo-electric field exist in the material, the electric polarization is related to the electric field by $P_{flexo} = \epsilon E_{flexo}$.

Reviewer #3 (Remarks to the Author):

The manuscript presents investigations of flexoelectricity at nanoscale and reports a significant enhancement of the flexoelectric effects. While the study itself is interesting, the advance made here is not sufficient to warrant the publication of Nature Communication. First, as Fig.3 shows, the STO becomes partially metallic as a function of strain gradient. However, the definition of the flexoelectric coefficient is not clear when a system becomes metallic. It is also known from previous studies that a composite material consisting of both metallic and insulating components may have a larger dielectric constant than the two constitutes, subsequently leading to large flexoelectric effects. So, what has been observed here may be very similar of what has been observed in the composite materials, by a strain gradient induced metal-insulator transition. Second, the SRO is metallic, how to separate the effects of SRO on the current study. Going back to the first point, the combination of SRO and STO may also lead to larger flexoelectric effects under certain conditions. A minor point about the DFT calculations, the connection between what has been simulated in Fig. 3 and experiments are rather vague. The atomistic configuration and displacement are artificial. Overall, I cannot recommend the publication of this manuscript.

Reviewer #1 (Remarks to the Author):

We would like to thank the reviewer for his/her in-depth reviews and excellent questions/suggestions regarding our manuscript. In the pages that follow, we provide our responses to each of the reviewer's questions, in order. The responses are written in blue.

The authors present an interesting study by combining flexoelectricity and quantum mechanical tunneling. Using a conductive AFM (atomic force microscope) the flexoelectric effect is induced and simultaneously the implication detected by the measured tunneling current.

We appreciate the reviewer for acknowledging the importance of our work.

Question #1

The experimental and theoretical issues have been carefully and thoughtfully done. Concerning mechanical related effects and quantum mechanical tunneling the authors overlooked the literature on the piezoelectric effect and electron transport in FTJs. Because the work the author present its novel, its necessary to include in the introduction also a paragraph and the piezo-effect on electron tunneling to complete the introduction. Here a number of references which should be cited?

Converse-piezoelectric effect on current-voltage characteristics of symmetric ferroelectric tunnel junctions. Xiaoyan Lu, Wenwu Cao, Wenhua Jiang, and Hui Li, J. Appl. Phys. 111, 014103 (2012); doi: 10.1063/1.3673600

Electroresistance Effect in Ferroelectric Tunnel Junctions with Symmetric Electrodes, Daniel I. Bilc et al., ASC Nano 2012

A ferroelectric tunnel junction based on the piezoelectric effect for non-volatile nanoferroelectric devices, Shuoguo Yuan et al., J. Mater. Chem. C, 2013, 1, 418

Theoretical current-voltage characteristics of ferroelectric tunnel junctions, H. Kohlstedt, N. A. Pertsev, J. Rodríguez Contreras, I and R. Waser, Phys. Rev. B 72, 125341 (2005)

Giant piezoelectric resistance in ferroelectric tunnel junctions, Yue Zheng and C H Woo Nanotechnology 20 (2009) 075401

Response #1

We thank the reviewer for his/her suggestion. We have included a separate paragraph in the introduction of the manuscript, citing the articles referred by the reviewer. In the revised manuscript, we now write:

“The electron tunnelling probability through a barrier mainly depends on two parameters: the barrier height and width. Modifying the barrier height and width, therefore, readily allows tuning the tunnelling current. This concept is used to realize ferroelectric tunnel devices^{18–20}, where tunnelling current is measured across an ultrathin ferroelectric layer sandwiched between metallic electrodes. The depolarization field, originating from the imperfect screening of ferroelectric polarization by the metallic electrodes, alters the intrinsic barrier height. For asymmetric electrodes, changing the polarization direction yields two different effective barrier heights, and subsequently leads to two discrete electroresistance states. Meanwhile, due to the converse piezoelectric effect, the barrier width can also modulate in response to the electric field applied during the current measurement^{21–25}. This also leads to dissimilar electroresistance states. In this article, we demonstrate that strain-gradient induced flexoelectric polarization allows systematically changing both the barrier height and width, enabling nanoscale characterization of flexoelectricity.”

Questions #2

In the supplements the authors confirm a stress free STO film by PFM. It would be more convincing to include a XRD (reciprocal space mapping) as well.

Response #2

We thank the reviewer for his/her suggestion. We have now performed the reciprocal space mapping around the STO (103) Bragg reflex, as shown in the figure below. Except the peaks from the STO substrate and bottom SRO layer, we did not detect any additional Bragg peak from the STO film. Therefore, this data suggests that our ultrathin STO barrier layer is strain-free. We have added this figure in the revised Supplementary Information.

[From Supplementary Fig. 14: Reciprocal space mapping]

Question #3

The effect on electron tunneling is attributed to the flexoelectric effect. The effect on strain on the SrRuO₃ was not taken into account. Can the authors indeed exclude any effect on the band structure of the SRO electrode due to the induced strain gradient? This might lead to a modified tunneling current.

Response #3

We thank the reviewer for raising this important question. First of all, as shown in the figure below, our phase-field simulations indicate that the strain gradient is much smaller in the SrRuO₃ (SRO) region (i.e., the region for $x_3 \leq 0$) than in the SrTiO₃ region. Furthermore, the strain-gradient-induced flexoelectricity is not well defined in metallic systems, such as SRO. Therefore, we will focus on the effect of strain itself in our discussion below.

[From Supplementary Fig. 4: Calculated strain gradients imposed by the AFM tip]

To understand how the strain affects the band structure of SRO and subsequently the tunnelling transport, we additionally performed first-principles DFT calculations. We fixed the in-plane lattice parameter of SRO to that of STO substrate, and imposed compressive strain u_{33} (ranging from 0 to -8%) in the out-of-plane direction. This assumption closely accounts for the strain distribution, obtained from the phase-field simulations (Supplementary Fig. 4). As shown in the figure below, our calculation suggests that with increasing the strain, the density of states at the Fermi energy (ρ_F) slightly increases and thus the screening length ($\delta_{\text{SRO}} \propto 1/\sqrt{\rho_F}$) could decrease, whereas the work function of SRO (W_{SRO}) slightly decreases by ~ 0.2 eV. Given the electrostatic constraint $\Delta\phi_1/\Delta\phi_2 = (\phi_{0,1} - \phi_1)/(\phi_2 - \phi_{0,2}) = \delta_{\text{SRO}}/\delta_{\text{PIR}}$, where $\phi_{0,1}$ is proportional to W_{SRO} , the influence of the decreased δ_{SRO} on the tunnel barriers seems to cancel out that of the decreased W_{SRO} . Furthermore, these changes in δ_{SRO} and W_{SRO} are too small to be responsible for the anomalous behavior of tunnel transports (as shown in the panel **c** below). Thus, we conclude that the effect of strain on SRO is not significant. To make this point clear, we revised our Supplementary Information by including new DFT data and related discussion.

[From Supplementary Fig. 11: **a**, Density of states of SRO for different out-of-plane strains u_{33} . **b**, Density of states at the Fermi energy (ρ_F ; red squares) and work function (W_{SRO} ; blue circles) of SRO as a function of u_{33} . **c**, Calculated tunneling I - V curves, corresponding to the tunnel barrier profiles (inset).]

Question #4

By calculating the I(V) characteristics the authors the electron mass m_e (see equation S1/suppl. material). It is unclear whether the authors use the free electron mass or an effective mass. This issue should be clarified.

Response #4

We have used the free electron mass in our calculation. In the Methods part, we clarified this by stating “Also, b , m_e , d , and $\phi_{1,2}$ are the baseline, free electron mass, barrier width, and barrier height, respectively.”

Reviewer #2:

We would like to thank the reviewer for his/her in-depth reviews and excellent questions/suggestions regarding our manuscript. In the pages that follow, we provide our responses to each of the reviewer's questions, in order. The responses are written in blue.

This manuscript by Saikat Das et al provides a really important method to characterize nanoscale flexoelectricity, and it is extremely important since strain gradient will be much stronger than that in macroscopic materials.

We appreciate the reviewer for acknowledging the importance of our work.

Question #1

The strain gradient is theoretically analyzed by the Hertz contact mechanics of the spherical indenter for elasticity or anisotropic materials, whether it is valid for flexoelectric materials need to be discussed.

Response #2

We thank the reviewer for raising this issue. The Hertz contact mechanics assumes a non-frictional contact between two isotropic, elastic materials. For flexoelectric materials, such as the incipient ferroelectric SrTiO₃ (STO) here, it is generally not linearly elastic because of the presence of electromechanical couplings (piezoelectric, flexoelectric, and electrostrictive effects), as well as antiferrodistortive-strain couplings (rotostrictive effect). To this point, we totally understand the reviewer's concern.

However, we use the Hertz contact mechanics only to obtain the stress distribution at the STO film surface. With this surface stress distribution as the top boundary condition (and zero displacements at the substrate bottom as the bottom boundary condition), we calculated the stress distribution in the whole system (the film and the substrate) by solving the mechanical equilibrium equation (Equation S3 in our Supplementary Information). In doing so, we also self-consistently take into account the electrostrictive coupling (thereby, piezoelectric effects), flexoelectric coupling, and rotostrictive coupling as eigenstrains (stress-free strains). The sought solution is analytical for thin films using the micro-elasticity theory and Stroh formalism, as thoroughly presented in our previous work [e.g., *Acta Materialia* **50**, 395–411 (2002)]. This approach allows us to reliably extend the Hertz contact mechanics to the

flexoelectric materials for obtaining stress/strain distribution under force imparted by the tip.

Following the reviewer's recommendation, we now include in the revised Supplementary Section 3 a discussion on the applicability of Hertz contact mechanics for flexoelectric materials.

Question #2

The transverse strain gradient $\partial u_t / \partial x_3 = \partial u_{11} / \partial x_3 + \partial u_{33} / \partial x_3$ is considered to represent the values of strain gradient, this may not be correct since when the P_3 component is measured in the experiments. There are only three independent flexoelectric coefficients for cubic SrTiO₃ materials, and $P_3 = \mu_{1133} \partial u_{11} / \partial x_3 + \mu_{3333} \partial u_{33} / \partial x_3$ the transverse strain gradient $\partial u_t / \partial x_3$ cannot represent the combination contribution of the flexoelectric effect.

Response #2

We totally agree with the reviewer's comment. In the manuscript, we defined the total transverse strain gradient as $\partial u_t / \partial x_3 = \partial u_{11} / \partial x_3 + \partial u_{22} / \partial x_3$, where u_{11} and u_{22} are the transverse strains (acting in the plane of the STO film). Meanwhile, we ignored the longitudinal strain gradient (i.e., $\partial u_{33} / \partial x_3$) associated with the longitudinal strain (u_{33}) acting along the film normal direction, because this contribution is an order of magnitude smaller than the total transverse strain gradient. For details, kindly see the fifth paragraph of the main text and the Supplementary Fig. S4.

Question #3

Flexoelectric effect will generate a built-in electric field and it can be represented as $E_{\text{flexo}} = f_{\text{eff}} \partial u_t / \partial x_3$, therefore, Eq. (1) may be rewritten as

$$(\varphi_2 - \varphi_1) / ed = E_{\text{flexo}} + E_{\text{bi}} = f_{\text{eff}} (\partial u_t / \partial x_3) + E_{\text{bi}}$$

The electric polarization induced by the strain gradient is $P_{\text{flexo}} = \mu_{\text{flexo}} (\partial u_t / \partial x_3)$ and when only the flexo-electric field exists in the material, the electric polarization is related to the electric field by $P_{\text{flexo}} = \epsilon E_{\text{flexo}}$.

Response #3

The flexoelectric effect polarizes the STO barrier layer in the presence of strain gradients, which can be expressed as $P_{\text{flexo}} = \mu_{\text{eff}} (\partial u / \partial x)$. Conceptually, this can be perceived as being the polarization response of the medium to an applied flexoelectric field $E_{\text{flexo}} = f_{\text{eff}} (\partial u / \partial x)$. It is, however, important to note that this flexoelectric field is not a real electric field that obeys Gauss's law in electrostatics, and can be thought of as a pseudo electric field [see, e.g., "P. V.

Yudin & A. K. Tagantsev, Nanotechnology **24**, 432001 (2013)"]. The real electric field that acts on the tunnelling barrier is the depolarization field, E_{depol} , associated with P_{flexo} . In ultrathin STO, one can write $E_{\text{depol}} \sim -P_{\text{flexo}}/\varepsilon = -f_{\text{eff}}(\partial u_i/\partial x)$, where the minus sign is to imply that the direction of E_{depol} is opposite to P_{flexo} (or E_{flexo}). To comply with the reviewer's comment, we added Supplementary Section 9 in the supplementary information and revised the paragraph in the main text as below:

“The transverse strain gradient polarizes the STO layer through the flexoelectric effect, and the induced polarization can be expressed as $P = \varepsilon f_{\text{eff}}(\partial u_i/\partial x_3)$, where ε and f_{eff} are the dielectric permittivity and the effective flexocoupling coefficient of STO, respectively. In ultrathin STO, this flexoelectric polarization results in depolarization field ($\propto -P/\varepsilon$) and modifies the tunnel barrier profile according to the following electrostatic equation (Supplementary Section 9)^{18,19}:

$$(\varphi_2 - \varphi_1)/ed = P/\varepsilon + E_{bi} = f_{\text{eff}}(\partial u_i/\partial x_3) + E_{bi}, \quad (1)$$

where E_{bi} is the additional built-in field contribution that could arise from the work function difference between SRO and PtIr, surface dipoles³¹, and/or an offset between the calculated and actual strain gradients.”

Reviewer #3 (Remarks to the Author):

We would like to thank the reviewer for his/her in-depth reviews and excellent questions/suggestions regarding our manuscript. In the pages that follow, we provide our responses to each of the reviewer's questions, in order. The responses are written in blue.

The manuscript presents investigations of flexoelectricity at nanoscale and reports a significant enhancement of the flexoelectric effects. While the study itself is interesting, ...

We thank the reviewer for finding our work interesting.

...the advance made here is not sufficient to warrant the publication of Nature Communication.

Here, we would like to summarize main achievements of our study. (1) We for the first time demonstrated how to characterize nanoscale flexoelectricity under giant strain gradients. This is extremely important for fully utilizing the nanoscale flexoelectricity, which will be much stronger and more functional than that in macroscopic materials. (2) For doing this, we combined quantum tunnelling with flexoelectricity, which is conceptually original. (3) Furthermore, as we will discuss in more detail below, we characterized a fundamental quantity, i.e., 'flexocoupling' coefficient that does not depend on a material's dielectric permittivity. (4) Last, we discovered the enhancement of nanoscale flexocoupling coefficient under large strain gradients. Based on these arguments, we believe that our study will greatly advance the understanding and researches on flexoelectricity.

Question #1

First, as Fig.3 shows, the STO becomes partially metallic as a function of strain gradient. However, the definition of the flexoelectric coefficient is not clear when a system becomes metallic.

Response #1

We agree with the reviewer's comment that when a system becomes metallic, its flexoelectricity is not well defined. However, please note that for estimating the flexocoupling coefficient, we considered only the regime (i.e., for $\partial u_i / \partial x_3 \leq (\partial u_i / \partial x_3)_c = 1.56 \times 10^7 \text{ m}^{-1}$), where the whole STO region remains insulating, as shown in Fig. 4 of the main text. That is,

we analysed the data only in the yellow regime of the panels **g** and **h** below, which corresponds to the tunnel-barrier profiles as in the panel **a** or **b**. Under these circumstances, therefore, the definition of the flexocoupling coefficient still remains valid.

[From Figure 2 in our manuscript]

Question #2

It is also known from previous studies that a composite material consisting of both metallic and insulating components may have a larger dielectric constant than the two constituents, subsequently leading to large flexoelectric effects. So, what has been observed here may be very similar of what has been observed in the composite materials, by a strain gradient induced metal-insulator transition.

Response #2

We thank the reviewer for raising this interesting question. Indeed, Li *et al.* have previously reported an enhanced flexoelectric effect in composite materials consisting of insulating and metallic components [Appl. Phys. Lett. **103**, 142909 (2013)]. They showed that the dielectric permittivity in the composite material can become larger than in the constituent components, which in turn enhances the effective flexoelectric coefficient μ_{eff} (in the unit of C/m). This is very reasonable, as μ_{eff} is known to linearly scale with dielectric permittivity, following the

relation $\mu_{\text{eff}} = \varepsilon \cdot f_{\text{eff}}$, where ε and f_{eff} are the dielectric permittivity and **flexocoupling coefficient** (in the unit of V), respectively [see, e.g., “P. Zubko *et al.*, *Annu. Rev. Mater. Res.* **43**, 387–421 (2013)”]. Importantly, the **flexocoupling coefficient** f_{eff} is independent of materials’ dielectric permittivity and therefore a more fundamental quantity [see, e.g., “M. Stengel, *Phys. Rev. B* **90**, 201112(R) (2014)”].

As previously mentioned in “Response #1”, however, we estimated the flexocoupling coefficient by considering only the regime, where the whole STO region remains insulating. In this case, the STO layer cannot be a composite system consisting of insulating and metallic components. Furthermore, our approach allows direct characterization of the effective **flexocoupling coefficient** f_{eff} (in the unit of V) that does not depend on the dielectric permittivity, so we revealed that this fundamental quantity could become larger at the reduced dimension. Considering the aforementioned arguments, therefore, we believe that our study is fundamentally distinct from any previous works on composite materials.

Question #3

Second, the SRO is metallic, how to separate the effects of SRO on the current study. Going back to the first point, the combination of SRO and STO may also lead to larger flexoelectric effects under certain conditions.

Response #3

As the reviewer rightly pointed out in “Question #1”, flexoelectricity is not well defined in metallic systems, so the metallic SRO itself cannot contribute to the observed flexoelectric effect. Instead, the dielectric property of STO might vary in our STO/SRO capacitor geometry, which could influence the flexoelectric coefficient $\mu (= \varepsilon \cdot f)$. Importantly, however, we directly characterized the **flexocoupling coefficient** f that does not depend on the dielectric permittivity. Furthermore, in those ultrathin STO capacitors, the dielectric permittivity can actually decrease due to the intrinsic size effect [see, e.g., “*Appl. Phys. Lett.* **89**, 242915 (2006)”]. Therefore, SRO cannot make a noticeable contribution to the observed enhancement in **flexocoupling coefficients**.

Question #4

A minor point about the DFT calculations, the connection between what has been simulated in Fig. 3 and experiments are rather vague. The atomistic configuration and displacement are artificial.

Response #4

While we do not know exactly how the atoms respond to the strain-gradient-induced flexoelectricity, Fig. 3 generally supports our idea of polarization-induced electronic band bending and the associated interfacial metallization. For more accurate DFT calculation taking into account actual atomistic configuration and displacement, it is necessary to directly image the atomic structure under tip-induced strain gradient with atomic resolution. This is very non-trivial and naturally, we have to leave it for a future study.

REVIEWERS' COMMENTS:

Reviewer #1 (Remarks to the Author):

Dear authors,
excellent work!

Reviewer #3 (Remarks to the Author):

Giving the author's response, my original assessment remains the same.

The authors highlighted some of the achievements of this studies in the reply, which actually demonstrate the lack of breakthrough needed to publications in Nature Comm.

- 1) It may be technologically relevant to characterize nanoscale flexoelectricity under giant strain gradients, the fact that nanoscale flexoelectricity can be much stronger than macroscopic flexoelectric effects is well known, due to the larger strain gradients at the nanoscale.
- 2) I don't quite understand what the author intends to express by stating "we combined quantum tunnelling with flexoelectricity".
- 3) There are also previous studies discussing the flexocoupling' coefficient and dielectric permittivity may not have a one-to-one correspondence, although in some of the materials they seem to be closely related.
- 4) It is also known that the flexocoupling coefficient is a function of strain gradients, due to second order effects. Not sure this is a significant discovery.

Summary: The investigation itself is certainly of interests to the flexoelectricity community. The concerns here are related to whether this study represents a significant breakthrough to our understanding of the flexoelectric effects, which is needed for a high impact publication.

REVIEWERS' COMMENTS:

We would like to thank the reviewers for reviewing our manuscript. In the pages that follow, we provide our responses to each of the reviewers' concerns, in order. The responses are written in blue.

Reviewer #1 (Remarks to the Author):

Dear authors,

excellent work!

We thank the reviewer for his/her compliment and reviewing our manuscript.

Reviewer #3 (Remarks to the Author):

Giving the author's response, my original assessment remains the same.

The authors highlighted some of the achievements of this studies in the reply, which actually demonstrate the lack of breakthrough needed to publications in Nature Comm.

In the following, we would like to argue that our achievement is a breakthrough in the understanding/advancement of nanoscale flexoelectricity.

1) It may be technologically relevant to characterize nanoscale flexoelectricity under giant strain gradients, the fact that nanoscale flexoelectricity can be much stronger than macroscopic flexoelectric effects is well known, due to the larger strain gradients at the nanoscale.

We thank the reviewer for agreeing that it is technologically relevant to characterize nanoscale flexoelectricity under giant strain gradients.

Here, we want to address why the characterization itself is a breakthrough. As mentioned by the reviewer, it has been generally believed that nanoscale flexoelectricity can be much stronger than macroscopic flexoelectric effect, due to larger strain gradients at the nanoscale. However, this relies on *a priori* assumption that flexocoupling coefficients would remain constant even in the

limit of giant strain gradients. Thus, it is necessarily important to estimate the nanoscale flexocoupling coefficients under giant strain gradients experimentally. Our work provides the first experimental demonstration of characterizing nanoscale flexocoupling coefficients under giant strain gradients. Through this experimental characterization, we furthermore highlight that flexoelectricity could become much more powerful at the nanoscale due to not only a larger strain gradient, but also an enhanced flexocoupling strength. Therefore, we believe that this is an important breakthrough for practical applications of nanoscale flexoelectricity.

In the Discussion section of our revised manuscript, we clarified this by stating, *“Such mechanical tunability allows experimentally determining the flexocoupling strength at the nanoscale, which we find to be much enhanced compared to that in bulk. This finding emphasizes that flexoelectricity could become much more powerful at reduced dimensions due to not only a large strain gradient, but also an enhanced coupling strength.”*

2) I don't quite understand what the author intends to express by stating “we combined quantum tunnelling with flexoelectricity”.

We are very sorry to confuse the reviewer with the phrase “we combined quantum tunnelling with flexoelectricity”. We intended to imply that the conventional way of characterizing direct flexocoupling coefficients involves measuring the displacement currents as a function of strain gradients. Unfortunately, this approach is not suitable for nanoscale measurement under giant strain gradients, and therefore requires a breakthrough. In this work, we propose and demonstrate a novel approach of characterizing the flexocoupling coefficients at the nanoscale under giant strain gradients, whereby we utilize the sensitivity of the quantum tunnelling to the strain gradient-induced flexoelectric polarization.

3) There are also previous studies discussing the flexocoupling' coefficient and dielectric permittivity may not have a one-to-one correspondence, although in some of the materials they seem to be closely related.

As correctly pointed out by the reviewer, it is well known that the flexocoupling coefficient and dielectric permittivity are basically independent of each other. In our manuscript, however, we

did not insist that this is our original finding. Instead, we use such well-known fact, in order to rebut the reviewer's previous comment: in his/her previous comments, the reviewer claimed that the observed enhancement of flexocoupling coefficient might come from a larger dielectric permittivity in our material system.

It seems that the reviewer may not distinguish “flexocoupling” coefficient from “flexoelectric” coefficient. To avoid further confusion, we specified the definition of both coefficients in our revised manuscript, by stating, “*the induced polarization can be expressed as $P = \mu_{\text{eff}} \cdot (\partial u_i / \partial x_3) = \varepsilon \cdot f_{\text{eff}} \cdot (\partial u_i / \partial x_3)$, where μ_{eff} , ε and f_{eff} are the effective flexoelectric coefficient, the dielectric permittivity and the effective flexocoupling coefficient of STO, respectively.*”

4) It is also known that the flexocoupling coefficient is a function of strain gradients, due to second order effects. Not sure this is a significant discovery.

Although higher-order flexoelectricity could make flexocoupling coefficient vary with strain gradients (especially for giant strain gradients), it has not been widely accepted due to lack of experimental confirmation. Our study for the first time provides direct experimental characterization of nanoscale flexoelectricity, which indeed supports the emergence of higher-order flexoelectricity under giant strain gradients. This experimental confirmation is a significant breakthrough, distinct from just a conjecture or theoretical prediction. Therefore, we believe that our finding brings a fresh perspective to the current understanding of nanoscale flexoelectricity, and could stimulate further theoretical/experimental studies.

Summary: The investigation itself is certainly of interests to the flexoelectricity community. The concerns here are related to whether this study represents a significant breakthrough to our understanding of the flexoelectric effects, which is needed for a high impact publication.

We thank the reviewer for finding our work to be of certain interests to the flexoelectricity community. As addressed above, our achievement, i.e., direct characterization of nanoscale flexoelectricity, should represent a significant breakthrough to the understanding of flexoelectricity.